# Ferroptosis in Cancer Progression

**DOI:** 10.3390/cells12141820

**Published:** 2023-07-10

**Authors:** Rongyu Zhang, Jinghong Chen, Saiyang Wang, Wenlong Zhang, Quan Zheng, Rong Cai

**Affiliations:** 1Department of Biochemistry & Molecular Cell Biology, Shanghai Jiao Tong University School of Medicine, Shanghai 200025, China; 19891653218@sjtu.edu.cn (R.Z.); celeborn_v@sjtu.edu.cn (J.C.); wangsaiyang@sjtu.du.cn (S.W.); i119710910059@sjtu.edu.cn (W.Z.); 2Center for Singl-Cell Omics, School of Public Health, Shanghai Jiao Tong University School of Medicine, Shanghai 200025, China

**Keywords:** ferroptosis, cancer, EMT, angiogenesis, metastasis

## Abstract

Ferroptosis is a newly discovered iron-dependent form of regulated cell death driven by phospholipid peroxidation and associated with processes including iron overload, lipid peroxidation, and dysfunction of cellular antioxidant systems. Ferroptosis is found to be closely related to many diseases, including cancer at every stage. Epithelial–mesenchymal transition (EMT) in malignant tumors that originate from epithelia promotes cancer-cell migration, invasion, and metastasis by disrupting cell–cell and cell–cell matrix junctions, cell polarity, etc. Recent studies have shown that ferroptosis appears to share multiple initiators and overlapping pathways with EMT in cancers and identify ferroptosis as a potential predictor of various cancer grades and prognoses. Cancer metastasis involves multiple steps, including local invasion of cancer cells, intravasation, survival in circulation, arrest at a distant organ site, extravasation and adaptation to foreign tissue microenvironments, angiogenesis, and the formation of “premetastatic niche”. Numerous studies have revealed that ferroptosis is closely associated with cancer metastasis. From the cellular perspective, ferroptosis has been implicated in the regulation of cancer metastasis. From the molecular perspective, the signaling pathways activated during the two events interweave. This review briefly introduces the mechanisms of ferroptosis and discusses how ferroptosis is involved in cancer progression, including EMT, cancer angiogenesis, invasion, and metastasis.

## 1. Ferroptosis

### 1.1. Brief Description of Iron Metabolism

Although only a trace element, iron plays an essential role in cellular metabolic processes. Iron is present in many forms in the diet and absorbed in the intestine. The most well-known way of absorption is in the form of nonheme iron [1]. Fe^3+^ in food is, at first, reduced to Fe^2+^ by duodenal cytochrome B and possibly other reducing enzymes and then crosses the brush border of enterocytes via divalent metal-ion transporter 1 (DMIT1). Enterocyte iron is exported to the blood via ferroportin 1 (FPN1) on the basolateral membrane. Ferroxidase hephaestin on the basolateral membrane oxidizes Fe^2+^ to Fe^3+^ so that it binds to plasma transferrin (TF) and is distributed throughout the body in the form of transferrin–iron complexes. When iron-loaded TF binds to transferrin receptor 1 (TfR1) on the cell plasma membrane, cells can take up iron by endocytosis. In addition, iron in diets can also be absorbed in the form of heme iron, mainly found in hemoglobin and myoglobin. However, the mechanism of heme iron absorption needs yet to be further investigated [2].

Iron uptake is essential for many key biological processes, such as oxygen transport, DNA replication, and mitochondrial function, etc. A large fraction of intracellular iron enters the mitochondria and is stored or used for the synthesis of heme and iron–sulfur clusters [3]. Heme is engaged in oxygen transport while iron–sulfur clusters are involved in electron transfer in the respiratory chain and act as cofactors for many other proteins required for critical cellular activities. The iron that is not utilized at once is stored in ferritin in an ion state and can be mobilized by ferritin phagocytosis when needed. On the whole, the intracellular iron content is regulated by the iron regulatory element–iron regulatory protein (IRE-IRP) system, while the whole-body iron supply is primarily regulated by the hepcidin–ferroportin (FPN) axis in response to the body’s iron requirements [4].

Iron is exerted from the body through the sloughing of intestinal epithelial cells, exfoliation of skin cells, and physiologic blood loss due to menstruation or minor trauma to epithelial linings. These are passive pathways that are not regulated by iron levels or factors related to iron homeostasis [5].

### 1.2. Mechanisms of Ferroptosis

#### 1.2.1. Iron Overload

The imbalance of iron metabolism homeostasis results in increased intracellular free iron, which is one of the important features of ferroptosis [6]. Although the specific mechanisms involved in ferroptosis are not yet clear, there is much evidence that ferrous iron plays the critical role in this process (Figure 1).

By using iron chelators, Dixon et al. successfully blocked ferroptosis both in vivo and in vitro. They also found that exogenous iron supplementation increased the sensitivity of cells to ferroptosis inducers [7]. Hou et al. observed an increase in cellular labile iron during the induction of ferroptosis [8]. Li et al. found that excess iron, both heme and nonheme iron, can directly induce ferroptosis [9]. In addition, excess iron produces reactive oxygen species (ROS) through the Fenton reaction and the Haber–Weiss reaction, triggering oxidative stress and promoting lipid peroxidation through nonenzymatic reactions [10].

Studies about iron metabolism in ferroptosis suggest that key molecules in the process of iron metabolism might all be regulators of cellular sensitivity to ferroptosis [11]. Given that drug-resistant cancer cells are vulnerable to ferroptosis and that a number of organ injuries and degenerative pathologies are driven by ferroptosis, therapeutic ideas targeting ferroptosis might offer a new turnaround for many diseases.

#### 1.2.2. Lipid Peroxidation

Polyunsaturated fatty acids (PUFAs) are susceptible to peroxidation during ferroptosis. The enzymes ACSL4 (Acyl-CoA synthetase long-chain family member 4) and LPCAT3 (lysophospholipid acyltransferase 3) are closely related to the synthesis of phospholipids containing polyunsaturated fatty acids (PUFA-PLs) (Figure 1). ACSL4 can cause the enrichment of long PUFAs in cell membranes [12]. LPCAT3 promotes the combination of PUFAs with phospholipids to form PUFA-PLs, which are susceptible to free-radicals-induced oxidation catalyzed by lipoxygenase.

Fatty acid synthesis mediated by acetyl-CoA carboxylase (ACAC) as well as release mediated by lipophagy induces the accumulation of intracellular free fatty acids, thereby accelerating ferroptosis. AMPK-mediated ACAC phosphorylation inhibits ferroptosis by preventing the synthesis of PUFAs [13].

Normally, several lipoxygenases participate in lipid peroxidation; for example, lipoxygenase-12/15 usually oxidizes PUFAs [14] while inhibition or knockdown of lipoxygenases inhibits ferroptosis in some cell types [15].

#### 1.2.3. Antioxidant Systems

The glutathione peroxidases (GPX) family is considered to be an important protective system in lipid peroxidation, among which GPX4 plays a key role in inhibiting ferroptosis by reducing phospholipid hydroperoxides to hydroxyphospholipid directly [16]. Drugs that inhibit GPX4 (class II ferroptosis-inducers), such as RSL3, ML162, ML210, can induce ferroptosis [17]. The containing of selenocysteine confers GPX4 increased antiferroptotic activity [18] and GSH as an essential cofactor, helps GPX4 reduce lipid hydroperoxide to lipid alcohols [19].

Cystine/glutamate transporter (also known as system Xc-) transports cystine (Cys2) into cells, which is then oxidized to cysteine (Cys). Cysteine is used to synthesize GSH under the catalysis of glutamate-cysteine ligase (GCL) and glutathione synthetase (GSS). System Xc- consists of two subunits, SLC7A11 (solute carrier family 7A member 11) and SLC3A2 (solute carrier family 3A member 11). The expression and activity of SLC7A11 are positively regulated by the transcription factor nuclear factor E2-related factor 2 (NRF2) [20] and negatively regulated by p53 [21], which helps to control the level of GSH in ferroptosis (Figure 1). System Xc- inhibitors (class I ferroptosis-inducers), including erastin, sulfasalazine, and sorafenib, can thus induce lipid peroxidation and ferroptosis [17].

The NADPH–FSP1–CoQ10 pathway is another important antioxidant pathway independent from the system Xc–GSH–GPX4 axis. CoQ10 is mainly synthesized in mitochondria and its reduced form, CoQ10H_2_, is a robust lipophilic antioxidant [22]. FSP1, ferroptosis inhibitor protein 1, is recruited to the plasma membrane to reduce CoQ10 to CoQ10H_2_, which effectively blocks the spread of lipid peroxides [23].

## 2. Ferroptosis and EMT in Cancer

Epithelial–mesenchymal transition (EMT) is the process by which epithelial cells lose specific epithelial phenotypes and acquire mesenchymal phenotypes and behaviors. The concept of EMT was first proposed during embryonic development [24] and subsequently observed in other processes including tissue repair, organ fibrosis, and cancer metastasis [25].

### 2.1. Characteristics and Mechanisms of EMT

Epithelial cells are connected by cell junctions, including adhesion junctions, tight junctions, gap junctions, and desmosomes. Cells are attached to the underlying basement membrane via hemidesmosomes and present apical–basal polarity. During the epithelial–mesenchymal transition, the cell junctions are substituted by cell–matrix junctions, and the apical–basal polarity is replaced by front–back polarity. The cytoskeleton is reprogrammed so that the cells turn into a spindle-shaped mesenchymal morphology and are conferred the ability of migration and invasion to the basement membrane [26]. Furthermore, hallmarks of cells are also greatly altered during EMT, mainly presented in the loss of epithelial markers and the acquisition of mesenchymal markers. E-cadherin, the epithelial marker, is downregulated while N-cadherin and vimentin, the mesenchymal markers, are upregulated during EMT [25].

EMT transcription factors (EMT-TFs) play a central role in the process of EMT. The major EMT-TFs include SNAI1/2, TWIST1/2, and ZEB1/2, which are involved in the repression of epithelial associated genes and the promotion of mesenchymal associated genes. SNAI1/2 and ZBE1/2 can directly bind to the E-box of the promoter of *CDH1*, which encodes E-cadherin, to repress its expression, while TWIST represses it indirectly [27]. SNAI1 and ZEB can downregulate the expression of genes that are related to the formation of tight junctions and apical–basal polarity. TWIST and ZEB1/2 can increase the expression of vimentin and N-cadherin and SNAI1 can act as a transcriptional activator to directly prompt mesenchymal gene expression. Alterations in the expression of these genes can affect cell–cell adhesion, cell polarity, and motility, resulting in the epithelial–mesenchymal transition of cells [28].

EMT-TFs could all be regulated by multiple miRNAs and post-translational modifications, including phosphorylation, ubiquitination, SUMOylation, acetylation, and histone acetytransferases(HATs) [28,29].

Many other factors have been reported to participate in EMT, including various growth factors, Wnt pathway, notch pathway, etc. For example, TGF-β interacts with the transcription factors, SNAI, ZEB, and TWIST, through the Smad pathway to affect the transcription of epithelial and mesenchymal-associated genes [30].

### 2.2. Ferroptosis and EMT in Cancer

EMT is divided into three distinct subtypes, which are involved in embryonic development [25], wound healing and tissue regeneration [31], and cancer metastasis, respectively. Type-III EMT is involved in the progression of cancer. As mentioned above, losing of cell–cell junctions and the altering of cell polarity during EMT endow the cancer cells with the ability to migrate, invade into adjacent tissues, intravasate into vascular or lymphatic vessels, and then enter the systemic circulation [26].

Some studies suggest that ferroptosis and EMT in cancer promote each other, forming a positive feedback loop. E-cadherin inhibits EMT by affecting cell–cell contacts while protecting cells from ferroptosis in experiments in vitro [32]. Enhancing the expression of EMT transcription factors such as SNAI, TWIST, and ZEB reverses the ferroptosis sensitivity of cells. Overexpression of ZEB1 increases cellular susceptibility to ferroptosis while silencing ZEB1 exhibits the opposite effect. Epigenetic reprogramming of EMT-TF, such as the elimination of CDH1 hypermethylation, promotes the expression of E-cadherin while decreased ferroptosis susceptibility [33]. Studies on human adrenal cortical carcinoma lines have found that sensitivity to ferroptosis was augmented due to the increased iron accumulation, ROS production, and decreased antioxidant genes in mesenchymal-like cells [34]. A strong correlation between ferroptosis and EMT was shown in RNA-seq and analysis of the TCGA dataset, with the high tendency of ferroptosis and EMT suggesting a poor prognosis of bladder cancer and lung adenocarcinoma [35,36]. Ferroptosis-related genes are used in the risk prognostic model of bladder cancer and colon adenocarcinoma [37]. While there is also research found that the EMT transcription factor slug, also known as SNAI2, which protects glioblastoma cells from ferroptosis by reversing the protein levels of SLC7A11, a subunit of system Xc-, which might contribute to the treatment of glioblastoma [38].

Ferroptosis is considered a regulator of EMT in the studies of various types of cancers. Inhibition of ferroptosis promotes epithelial–mesenchymal transition of cancer cells, leading to cancer migration, metastasis, and poor prognosis. Both ferroptosis inhibition and EMT induction occur in IRF2 overexpressed glioma cells and ARNTL2 overexpressed lung adenocarcinoma cells, which also suggests poor prognosis for both kinds of cancer [39]. Mechanistically, some processes related to ferroptosis such as ferritinophagy and ROS production were found to be involved in drug-induced EMT inhibition in hepatocellular carcinoma [40] and gastric cancer cell lines [41]; the inhibition of EMT was reversed with the Fer-1, a ferroptosis inhibitor. CST1 inhibits ferroptosis by increasing the stability of GPX4, a key protein of ferroptosis, and reducing intracellular reactive oxygen species, thereby promoting migration and invasion via EMT in gastric cancer cells [42]. In turn, the induction of ferroptosis prevents cancer cells from EMT and inhibits cancer progression. NRF2 is an essential regulator of the cellular antioxidant system. Normally located in the cytoplasm, NRF2 is translocated to the nucleus and functions upon oxidative stress [8]. NRF2 expression is controlled mainly by Keap1 through proteasomal degradation [43]. The activation of the Keap1–NRF2 pathway facilitates the inhibition of EMT, a process that may act through the induction of ferroptosis [41,44]. Inhibition of the PI3K–AKT signaling pathway induces ferroptosis in KLF2 overexpressed colorectal cancer(CRC) cells, which also results in EMT suppression of CRC cells and ultimately inhibits progression and metastasis of CRC [45].

In summary, iron plays an important regulatory role both in EMT and ferroptosis and there is a complicated regulatory network between ferroptosis and EMT in cancer, which exhibits different interactions in specific contexts.

## 3. Ferroptosis and Cancer Invasion and Metastasis

### 3.1. Cancer Angiogenesis, Invasion, and Metastasis

#### 3.1.1. Cancer Angiogenesis

Angiogenesis is the process of growing new blood vessels from existing ones and includes degradation of the vascular basement membrane, activation, proliferation, and migration of endothelial cells (ECs), and the generation of new blood vessels. Angiogenesis is one of the characteristics of cancer development [46]. Under normal physiological conditions, the vascular system remains quiescent for a long time after the maturation of the organism with transient angiogenesis occurring only when the organism is damaged. Angiogenesis is essential for cancer growth and metastasis because, by constant angiogenesis, the oxygen and nutrient demands of tumor cells are met [47].

The relationship between tumor growth and angiogenesis was investigated by Folkman, who found that when the blood supply to the tumor is cut off, the tumor shrinks or regresses due to ischemia [48]. Later, Dhar et al. found significantly higher levels of angiogenesis in differentiated thyroid cancer [49] and Mohammed et al. found that high levels of angiogenesis were closely associated with metastasis and recurrence as well as high mortality in colorectal cancer [50]. In addition, several independent research groups examined angiogenesis levels in various tumor samples using immunohistochemical techniques and found that angiogenesis was significantly upregulated in several common malignancies.

As research has progressed, the understanding of the mechanisms of cancer angiogenesis-related signaling pathways and their effects on cancer growth, metastasis, and invasion has increased. Two types of signaling pathways have been extensively studied, namely signaling pathways that regulate vascular permeability and those involved in vascular remodeling and vascular maturation. Among the signaling pathways that regulate vascular permeability, the most important is the VEGF–VEGFR signaling pathway. Among the regulatory pathways involved in vascular remodeling and maturation, the angiopoietin–Tie-2 signaling pathway is the most important. In addition, several other signaling pathways associated with angiogenesis are also involved in the regulation of vascular permeability and/or maturation. These signaling pathways are interrelated and interacted with each other to promote the proliferation of vascular ECs and induce tumor angiogenesis, thus providing favorable conditions for malignant tumor proliferation, invasion, and metastasis [51].

#### 3.1.2. Process of Cancer Invasion and Metastasis

Cancer metastasis is a multistep process that involves local invasion of cancer cells, intravasation, survival in circulation, arrest at a distant organ site, extravasation, and adaptation to foreign-tissue microenvironments [52].

Mutations in oncogenes may be one of the initial leading factors in this process [53]. Active protein hydrolysis by the action of MMPs (matrix metalloproteinases) can alter the tumor microenvironment and promote the absence of the basement membrane barrier, providing suitable conditions for cancer cell invasion. After crossing the basement membrane, cancer cells enter the stroma and interact with tumor-associated stromal cells [54]. The stimulation of cancer cells makes stroma “reactive”, while cytokines and growth factors, such as interleukin-6 (IL-6) and EGF (epidermal growth factor), secreted by stromal cells, enhance the malignant traits of cancer cells [55,56].

Molecular changes allow cancer cells to cross the pericytes and endothelial cell barriers, promoting intravasation [57]. As mentioned above, cancer cells stimulate the formation of new blood vessels within their microenvironment. These new blood vessels are more tortuous and prone to leakiness compared to normal blood vessels, which creates favorable conditions for tumor cells to enter the circulation [58].

In order to reach distant organ sites, circulating tumor cells (CTCs) must cope with a certain amount of survival stresses [57]. Studies have shown that platelets and their activation play a crucial role in the spread of tumor cells. Platelets in circulation are cross-linked by adhesion molecules and bind fibrinogen to form an additional immune escape layer that can protect tumor cells from killing by NK cells [59,60].

Tumor cells that survive successfully in circulation can theoretically disseminate throughout the body but, in practice, certain types of tumors tend to metastasize to specific organs. Filder et al. explain the distant metastasis of tumor cells with the “seed and soil” hypothesis [61]. They believe that successful metastasis depends on the interaction between metastatic cells and local homeostatic mechanisms. Another hypothesis suggests that CTCs themselves have a certain preference. Some cancer cells are able to adhere to specific tissues [62].

After being arrested by a distant organ, CTCs may begin to proliferate intraluminally and form a microcolony that eventually breaks the surrounding vessel wall [63]. In addition, cancer cells may penetrate the endothelial and pericyte layers, entering the tissue parenchyma directly. This process is known as extravasation. Primary tumors express proteins such as secreted protein angiopoietin-like-4 (Angptl4) and pleiotropically acting factors COX-2 and MMP to disrupt the microenvironment of distant metastatic organs and increase permeability [64,65].

After reaching the target organ site, tumor cells need to adapt to the new microenvironment and colonize there. Some researchers believe that the “premetastatic niche” formed at the target site by invading bone-marrow-derived cells (BMDCs) are involved in this progress [66]. Erler et al. demonstrated that lysyl oxidase (LOX) secreted by hypoxic breast tumor cells is able to accumulate at the target site and cross-link collagen IV, thereby recruiting CD11b^+^ cells. CD11b^+^ cells produce matrix metalloproteinase-2 (MMP2), enhancing the invasion and recruitment of BMDCs and promoting cancer metastasis [67].

The process of the invasion-transfer cascade above has been widely recognized. Metastasis cannot be successfully completed if any step of the process goes wrong. Apart from that, various factors such as lncRNA and EGR1 have been found to play a role in cancer invasion and metastasis [68,69,70].

### 3.2. Relationship between Ferroptosis and Cancer Angiogenesis, Invasion, and Metastasis

Since ferroptosis was formally defined in 2012 [71], studies regarding ferroptosis and tumor biological behaviors have attracted a lot of interest (Figure 2). Studies have shown that there exist crosstalk between ferroptosis and tumor-associated signaling pathways, including RAS, TP53, NFE2L2, HIF pathways, and so on, which serves as a basis for investigating new targets for tumor therapy [72].

#### 3.2.1. Factors That Promote Tumor Angiogenesis May Inhibit Ferroptosis

Under certain conditions, hypoxia induces angiogenesis, which is controlled by a family of hypoxia inducible factors (HIFs), including HIF-1α and HIF-2α [73]. HIF-1α instability in nonsmall cell lung cancer cells has been reported to reduce the susceptibility to ferroptosis [74]. In addition, HIF-1α knockdown or HIF-2α ablation in renal clear cell carcinoma cells alone was found to reduce the susceptibility of tumor cells to ferroptosis [75]. Thus, the HIF pathway may be a major driver of ferroptosis susceptibility in several tumors and is essential for promoting ferroptosis in tumor cells.

Most of the factors that promote tumor angiogenesis play a role in inhibiting tumor ferroptosis. Exosomes accelerate tumor growth by participating in neovascularization. In RSL3-induced ferroptosis in breast cancer cells, exosome secretion is a protective mechanism against ferroptosis [76]. Exosomal miR-522 secreted by CAFs also inhibits ferroptosis in gastric cancer [77]. Integrins not only promote tumor metastasis but also contribute to tumor angiogenesis. Integrin α6β4 protects breast cancer cell-adherent epithelial cells and cancer cells from erastin-induced ferroptosis [78] and integrins have an inhibitory role in driving the ferroptosis pathway in breast cancer cells.

#### 3.2.2. Ferroptosis Affects Different Steps of Cancer Invasion and Metastasis

First of all, as a special form of cell death, ferroptosis somehow determines the survival and growth of cancer cells (Figure 2). Only by escaping from ferroptosis can cancer cells undergo the process that follows. A research project conducted by Chen et al. revealed that the anticancer activity of erianin is associated with ferroptosis and cancer cell proliferation and migration [79]. Ferroptotic events, including ROS accumulation, GSH depletion, and lipid peroxidation, were significantly triggered following treatment with erianin. More importantly, pretreatment with the ferroptosis inhibitor Fer-1, Lip-1, or DFO reduced erianin-induced cell death and suppressed cell migration. Their hypothesis, erianin triggers ferroptosis through the Ca^2+^/CaM signaling pathway, was preliminarily confirmed. They also proved that erianin inhibited cell proliferation by G2/M-phase arresting. Nassar et al. investigated that *DECR1*, encoding the rate-limiting enzyme for oxidation of PUFAs, is highly expressed in prostate cancer (PCa) tissues [80]. *DECR1* knockdown selectively inhibits β-oxidation of PUFAs, enhances oxidative stress and lipid peroxidation in mitochondria [81], and ultimately leads to ferroptosis. Nassar et al. suggested that the consistently increased expression of *DECR1* in PCa tissue might be a solution to avoid cell death and, therefore, contributes to PCa cell viability and invasive behavior [80].

Then, there is some evidence for the effect of ferroptosis on cancer invasion. You et al. constructed a scoring system based on ferroptosis-related genes to investigate the relationship between ferroptosis and clinical features of ovarian cancer [82]. The results showed that a cluster with higher ferroptosis-resistant-related genes has shorter median survival times. Critically, this cluster also exhibited aggressive growth patterns, including blood infiltration and lymphatic infiltration, suggesting that ferroptosis may affect ovarian cancer progression by regulating invasion ability.

Cancer cells are still affected by ferroptosis after entering the blood or lymph. Ubellacker et al. conducted a study on melanoma metastasis [83]. The results showed that melanoma cells in lymph experience less oxidative stress and form more metastasis than those in blood. Differences in lymph and plasma, such as higher levels of glutathione and oleic acid and lower levels of free iron, help reduce ferroptosis in melanoma cells in the lymph. They also tentatively demonstrated that the function of oleic acid in protecting melanoma cells from ferroptosis is associated with acyl-CoA synthetase long-chain family member 3 (ACSL3). In their study regarding the CTCs of melanoma, Hong et al. demonstrated that TF may be involved in the regulation of blood-borne metastasis of melanoma [84]. TF downregulation reduces intracellular iron pools, ROS, and lipid peroxidation, conferring resistance to ferroptosis inducers. These changes in CTCs of melanoma can be induced by the lipogenic regulator SREBP2, while the tumorigenic defects of endogenous *TF* knockdown can be partially rescued by the lipophilic antioxidants ferrostatin-1 and vitamin E. That is how SREBP2-driven iron homeostatic pathways contribute to cancer progression.

Overall, there are many factors involved in both the regulation of ferroptosis and cancer progression. Lu et al. found that ACADSB, an acyl-CoA dehydrogenase localized in the mitochondria and nucleus, was downregulated in colorectal cancer tissues. They demonstrated that the overexpression of ACADSB inhibits CRC cell migration, invasion, and proliferation while promoting ferroptosis [85]. In other words, ACADSB may exert suppressive functions against colorectal cancer, which is consistent with the previously identified role of ACADSB in poor-differentiated hepatocellular carcinoma and renal clear cell carcinoma [86,87]. The mechanism of this function might be that ACADSB can promote the lipid metabolism via catalyzing the dehydrogenation of acyl-CoA derivatives [88]. Zhang et al. discovered that circRHOT1 contributed to invasion and metastasis and attenuated ferroptosis in breast cancer by regulating the miR-106a-5p/STAT3 (signal transducer and activator of transcription 3) axis [89]. Cao et al. showed that glucose-6-phosphate dehydrogenase (G6PD) can promote proliferation, migration, and invasion of hepatocellular carcinoma (HCC) and inhibit ferroptosis by downregulating POR (cytochrome P450 oxidoreductase) [90]. ACSL4, an important molecule in ferroptosis, is shown to be associated with migration, proliferation, and invasion of 17β-estradiol-induced cancers [91].

All these studies have suggested an inextricable link between ferroptosis and cancer invasion and metastasis but further studies are necessary to clarify the specific and comprehensive mechanisms.

#### 3.2.3. Pathways Associated with Both Iron Death and Tumor Progression

The importance of hypoxia-inducible factor-1 (HIF-1) in tumor development has long been demonstrated [92]. Studies in recent years have shown that HIF is involved in regulating the expression of genes related to lipid metabolism, such as *SCD1* and fatty-acid desaturase 2 (*FADS2*), thus affecting ferroptosis [74]. In addition, HIF-2 activates the expression of hypoxia inducible lipid droplet-associated protein (HILPDA), which induces a ferroptosis-susceptible cell state [75].

While p53-mediated cell-cycle arrest, senescence, and apoptosis are critical barriers to cancer development, P53 also plays an important role in cellular metabolism, oxidative responses, and ferroptosis [93]. P53 can both enhance ferroptosis by promoting the accumulation of lipid hydroperoxides and suppress ferroptosis by decreasing the accumulation of lethal lipid peroxides (LPO) [93,94,95].

Xu et al. confirmed that SLC7A11 could promote the proliferation, migration, and invasion of renal cancer cells (RCCs) by enhancing GPX4 output, which in turn inhibits ferroptosis [96].

A recent study conducted by Wang et al. showed that RB1 loss/E2F activation sensitized cancer cells to ferroptosis by upregulating the expression of ACSL4 and enriching ACSL4-dependent arachidonic acid-containing phospholipids [97]. Upon RB1 loss, E2F transcription factors induce ACSL4 expression, resulting in PUFA-PL accumulation and a high ferroptotic potential, which is kept in check by GPX4. This is consistent with the fact that GPX4 is abundantly expressed in most cancer cells. However, this strong ferroptotic potential is unblocked when RB1-deficient cells are treated with GPX4 inhibitors, resulting in massive ferroptosis [98].

## 4. Perspective

Since the discovery of ferroptosis as a novel regulated cell death, extensive attention has been drawn to investigate it, resulting in a more comprehensive understanding of iron metabolic processes and ferroptosis mechanisms.

Recent studies have revealed close links between ferroptosis and tumor malignancy, which propose that the expression of ferroptosis-associated genes could serve as prognostic indicators for cancer patients and that targeting ferroptosis may be a potential strategy to prevent cancer metastasis and drug resistance. Small molecules-induced ferroptosis has been found to enhance the sensitivity of chemotherapeutic drugs, especially in the condition of drug resistance [99]. Additional studies have demonstrated that ferroptosis inducers may be able to impede tumor infiltration and metastasis. For instance, Chen et al. confirmed that erianin could inhibit cell proliferation and suppress migration in lung cancer cells through triggering ferroptosis [79]. Nonetheless, histological and pharmacological studies are needed to assess the potential toxic effects, and the optimal drug dosing and scheduling remain to be explored. Large-scale, multicenter collaborative clinical trials are also needed before the clinical applications of such therapy. Furthermore, we have not yet explored the panoramic view of the relationship between ferroptosis and tumor cell invasion and infiltration. As the relevant mechanisms are further explored in the future, new targets will provide more possibilities for cancer treatment. Thus, the application of tumor therapy based on the molecular regulation mechanism of ferroptosis has a great research prospect and development potential.

## Figures and Tables

**Figure 1 cells-12-01820-f001:**
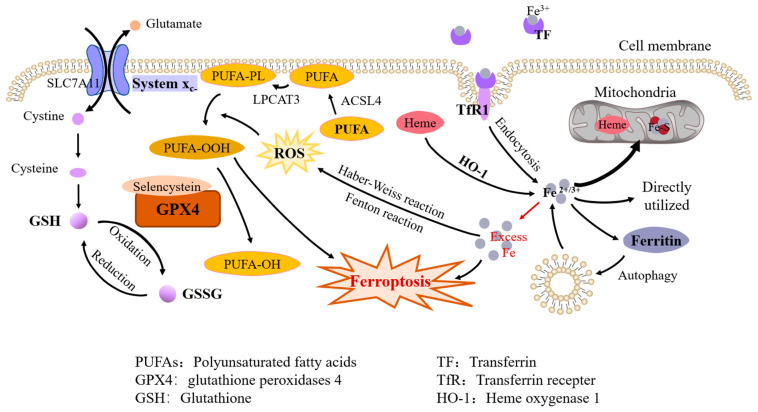
Molecular mechanisms of ferroptosis. Iron metabolism. Fe^3+^ is bound to TF and transported in plasma and taken up by endocytosis when iron-loaded TF binds to TfR1 on the cell membrane; intracellular heme and ferritin degradation are also sources of free iron. Intracellular iron is used to synthesize heme and iron–sulfur clusters in the mitochondria, or directly utilized in the cytoplasm, with excess iron being stored in ferritin. Iron overload. Excess iron can lead to ferroptosis directly and it also produces ROS through the Fenton reaction and the Haber–Weiss reaction, which promotes the peroxidation of PUFAs on the cell membrane. Antioxidant system. GPX4 can inhibit ferroptosis by reducing lipid peroxidation and GPX4 containing selenocysteine exhibits increased antiferroptosis activity. GSH is a cofactor of GPX4. Cysteine, an essential constituent of GSH, is transported into cells by system Xc-, which consists of two subunits, SLC7A11 and SLC3A2.

**Figure 2 cells-12-01820-f002:**
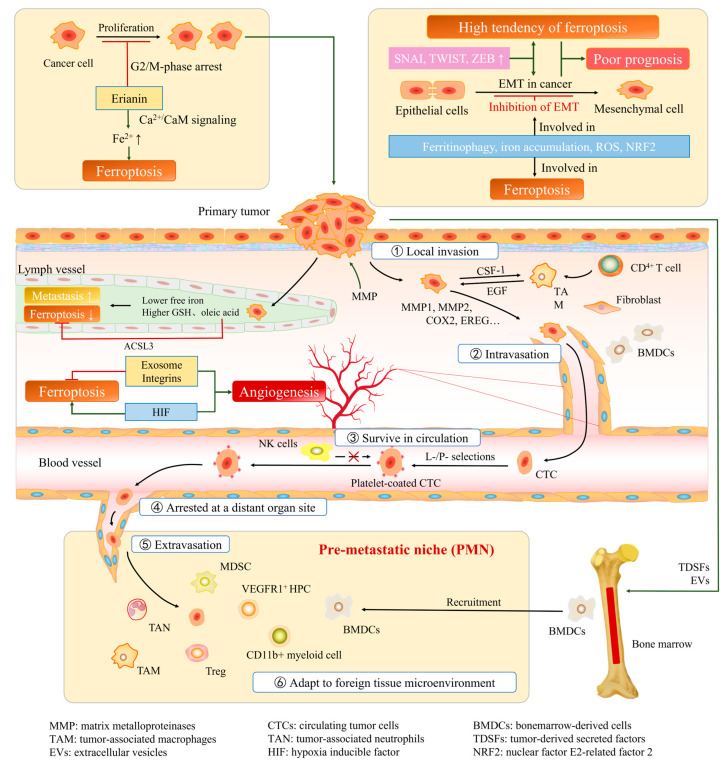
Ferroptosis in cancer progression. Erianin can inhibit cancer cell proliferation by G2/M phase arresting, and trigger ferroptosis. Enhanced expression of EMT-TFs reverses the ferroptosis sensitivity of cancer cells. Both the high tendency of ferroptosis and EMT suggest a poor prognosis of cancer. Processes characteristic of ferroptosis, such as ferritinophagy and ROS accumulation, are also found in drug induced EMT inhibition. MMPs can alter the tumor microenvironment and promote the absence of the basement membrane barrier, which makes the primary tumor more likely to enter the stroma and interact with tumor-associated stromal cells. Cytokines and growth factors are secreted by stromal cells and enhance the malignant traits of cancer cells. Cancer cells stimulate the formation of new blood vessels, which are more tortuous and prone to leakiness, and liable for cancer cells to enter the circulation. CTCs are coated by platelets to form an immune escape layer, which protects cancer cells from killing by NK cells. After reaching distant tissues, cancer cells penetrate the endothelial and pericyte layers to enter the target organ. To adapt to the new microenvironment, BMDCs are recruited and help to form the “premetastatic niche”. Hypoxia induces angiogenesis, which is controlled by HIFs, while the HIF pathway promotes ferroptosis susceptibility in several tumors. Exosomes and integrins both play roles in promoting angiogenesis and inhibiting ferroptosis. Higher levels of GSH and oleic acid and a lower level of free iron in lymph help reduce the ferroptosis of cancer cells, and that is why cancer cells in the lymph form more metastasis than those in blood.

## Data Availability

Not applicable.

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
