# Peer review of "Ferroptosis in Cancer Progression"

_cells, 2023, doi:10.3390/cells12141820_

Round 1

Reviewer 1 Report

Dear Authors,

thanks for submitting this review.

- It is well written, but imo it is not very strigently designed. You use a whole chapter to describe different steps of metastasis followed by only one chapter 3.2 in which you tell us about single findings in different systems where ferroptosis might play a role in invasion, migration etc. I kindly ask you to sort the information according to the different steps and provide sub headings.

- For CTCs please refer to PMID: 33203734

- After having finished reading the manuscript I was askig myself "and now? what is new? or is there any hypothesis the authors are drawing? Please provide some otherwise this review is only a collection of data.

- Please provide proper self-explanatory figure legends

- Resolution of Fig 2 is insufficient

Thanks

Few spelling mistakes

Author Response

Thank you for your constructive and kind suggestions.

Our responses to your comments:

  1. It is well written, but imo it is not very strigently designed. You use a whole chapter to describe different steps of metastasis followed by only one chapter 3.2 in which you tell us about single findings in different systems where ferroptosis might play a role in invasion, migration etc. I kindly ask you to sort the information according to the different steps and provide sub headings.

Response: Thanks. We did the revision as you suggested.

  1. For CTCs please refer to PMID: 33203734

Response: Thank you for kindly providing us the reference. We did the citation as Ref 84.

  1. After having finished reading the manuscript I was askig myself "and now? what is new? or is there any hypothesis the authors are drawing? Please provide some otherwise this review is only a collection of data.

Response: Thank you for the question. In PERSPECTIVE, we supplied that what we should do in the future by taking ferroptosis induction for cancer treatment in the clinic, which is a new opinion for the readers to be think about.  

  1. Please provide proper self-explanatory figure legends

Response: We added the explanation in details in figure legends, thank you for the remind.

  1. Resolution of Fig 2 is insufficient

Response: We improved the quality of Fig 2.

Reviewer 2 Report

In their review, Zhang R et al. reported the actual literature about the possible link between mechanisms of ferroptosis and cancer progression, including epithelial-mesenchymal transition (EMT), cancer angiogenesis, invasion and metastasis.

This topic has some merit because ferroptosis is a new field of iron metabolism and the potential association between this pathomechanism and cancer progression is of great interest for clinicians as well as researchers.

The review is clear, concise and well written and not overloaded with tables and figures.

Before publication, some minor points should be considered:

       Page 3, line 114: The second headline “2. FERROPTOSIS AND AND EMT IN CANCER”should be written in bold.

·         Conclusion, page 9, line 379: “Additional studies have demonstrated that…” instead of “Additional studies have demonstrate that…”

Author Response

Thank you for your nice suggestions.

Our responses to your comments:

  1. Page 3, line 114: The second headline “2. FERROPTOSIS AND AND EMT IN CANCER”should be written in bold.

Response: We did the correction as you suggested, thanks.

  1. Conclusion, page 9, line 379: “Additional studies have demonstrated that…” instead of “Additional studies have demonstrate that…”

Response: We did the correction as you mentioned, thank you for the kind remind.

Round 2

Reviewer 1 Report

Dear Authors,

thanks for your positive reply and amendments

All the best